# MEGALODON: Efficient LLM Pretraining and Inference with Unlimited Context Length

**Xuezhe Ma**[π][*] **Xiaomeng Yang**[μ][*] **Wenhan Xiong**[μ] **Beidi Chen**[κ] **Lili Yu**[μ]

**Hao Zhang**[δ] **Jonathan May**[π] **Luke Zettlemoyer**[μ] **Omer Levy**[μ] **Chunting Zhou**[μ][*]

[μ]AI at Meta        [π]University of Southern California
[κ]Carnegie Mellon University        [δ]University of California San Diego

## Abstract

The quadratic complexity and weak length extrapolation of Transformers limits their ability to scale to long sequences, and while sub-quadratic solutions like linear attention and state space models exist, they empirically underperform Transformers in pretraining efficiency and downstream task accuracy. We introduce MEGALODON, an neural architecture for efficient sequence modeling with unlimited context length. MEGALODON inherits the architecture of MEGA (exponential moving average with gated attention), and further introduces multiple technical components to improve its capability and stability, including *complex exponential moving average (CEMA)*, *timestep normalization* layer, *normalized attention* mechanism and *pre-norm with two-hop residual* configuration. In a controlled head-to-head comparison with LLAMA2, MEGALODON achieves better efficiency than Transformer in the scale of 7 billion parameters and 2 trillion training tokens. MEGALODON reaches a training loss of 1.70, landing mid-way between LLAMA2-7B (1.75) and 13B (1.67). The improvements of MEGALODON over Transformers are robust throughout a range of benchmarks across different tasks and modalities. **Code**: https://github.com/XuezheMax/megalodon

## 1 Introduction

In many real-world applications, such as multi-turn conversation, long-document comprehension, and video generation, large language models (LLMs) must efficiently process long sequential data, understand internal long-range dynamics, and generate coherent output. The Transformer architecture (Vaswani et al., 2017), despite its remarkable capabilities, faces challenges with quadratic computational complexity and limited inductive bias for length generalization, making it inefficient for long sequence modeling (Wang et al., 2024; Zhou et al., 2024). Even with recently proposed distributed attention solutions (Li et al., 2023b; Liu et al., 2024), computing a single training step of a 7B parameter model over a 1M-token sequence is more than 100 times slower than performing the equivalent computation using 256 separate sequences of 4K tokens each.

Techniques like efficient attention mechanisms (Tay et al., 2020; Ma et al., 2021) and structured state space models (Gu et al., 2022a; Poli et al., 2023; Gu and Dao, 2023) have been introduced to overcome these limitations, aiming to enhance scalability and performance. However, the practical application of these methods still falls short of Transformers (Tay et al., 2022; Gu and Dao, 2023). This work introduces an unlimited context model that outperforms the canonical Transformer architecture on real-world language modeling.

---

[*]Equal Contribution. Correspondence to chuntinz@meta.com

38th Conference on Neural Information Processing Systems (NeurIPS 2024).

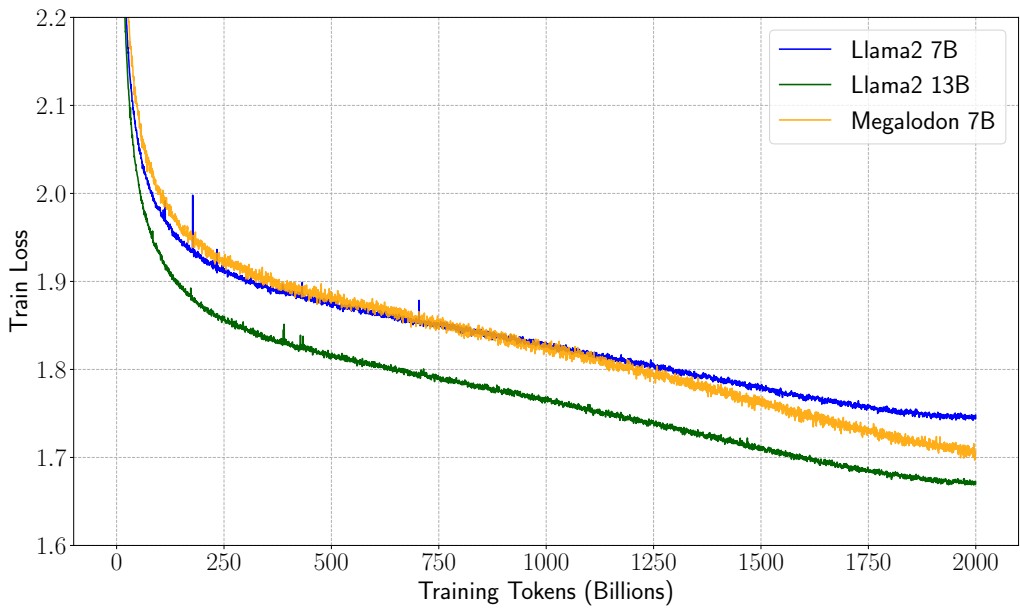

Figure 1: **Negative log-likelihood** for MEGALODON-7B, LLAMA2-7B and LLAMA2-13B.

Table 1: **Performance on standard academic benchmarks**, compared to open-source base models. We reported model size, context length and total data tokens during model pretraining. – indicates that the number was not reported in the original paper.

| Model | Size | Tokens | Context | MMLU | BoolQ | HellaSw | PIQA | SIQA | WinoG | Arc-e | Arc-c | NQ | TQA |
|---|---|---|---|---|---|---|---|---|---|---|---|---|---|
| Mamba | 3B | 0.6T | 2K | 26.2 | 71.0 | 71.0 | 78.1 | – | 65.9 | 68.2 | 41.7 | – | – |
| RWKV | 7B | 1.1T | 4K | – | – | 70.8 | 77.3 | – | 68.4 | 74.9 | 46.1 | – | – |
| MPT | 7B | 1T | 4K | 26.8 | 75.0 | 76.4 | 80.6 | 48.5 | 68.3 | 70.2 | 42.6 | 20.8 | 50.4 |
| Mistral | 7B | – | 16K | 60.1 | **83.2** | **81.3** | **82.2** | 47.0 | **74.2** | 80.0 | **54.9** | 23.2 | 62.5 |
| Gemma | 8B | 6T | 8K | **64.3** | **83.2** | 81.2 | 81.2 | **51.8** | 72.3 | **81.5** | 53.2 | 23.0 | 63.4 |
| LLAMA2 | 13B | 2T | 4K | 54.8 | 81.7 | 80.7 | 80.5 | 50.3 | 72.8 | 77.3 | 49.4 | **31.2** | **65.1** |
| LLAMA2 | 7B | 2T | 4K | 45.3 | 77.4 | 77.2 | 78.8 | 48.3 | 69.2 | 75.2 | 45.9 | 25.7 | 58.5 |
| MEGALODON | 7B | 2T | 32K | 49.8 | 80.5 | 77.5 | 80.1 | 49.6 | 71.4 | 79.8 | 53.1 | 25.7 | 60.5 |

We introduce MEGALODON, an improved MEGA architecture (Ma et al., 2023), which harnesses the gated attention mechanism with the classical exponential moving average (EMA) (Hunter, 1986) approach (§2). To further improve the capability and efficiency of MEGALODON on large-scale long-context pretraining, we propose multiple novel technical components. First, MEGALODON introduces the *complex exponential moving average (CEMA)* component, which extends the multi-dimensional damped EMA in MEGA to the complex domain (§3.1). Then, MEGALODON proposes the *timestep normalization* layer, which generalizes the group normalization layer (Wu and He, 2018) to autoregressive sequence modeling tasks to allow normalization along the sequential dimension (§3.2). To improve large-scale pretraining stability, MEGALODON further proposes *normalized attention* (§3.3), together with *pre-norm with two-hop residual* configuration by modifying the widely-adopted pre- and post-normalization methods (§3.4). By simply chunking input sequences into fixed blocks, as is done in MEGA-chunk (Ma et al., 2023), MEGALODON achieves linear computational and memory complexity in both model training and inference.

Empirically, we demonstrate the potential of MEGALODON as a general architecture for modeling long sequences, by evaluating its performance across multiple scales of language modeling, as well as downstream domain-specific tasks. Through a direct comparison with LLAMA2, while controlling for data and compute, MEGALODON-7B significantly outperforms the state-of-the-art variant of Transformer used to train LLAMA2-7B (Touvron et al., 2023) on both training perplexity (Figure 1) and across downstream benchmarks (Table 1). Evaluation on long-context modeling, including perplexity in various context lengths up to 2M and long-context QA tasks in Scrolls (Parisotto et al., 2020) prove MEGALODON's ability to model sequences of unlimited length. Additional experimental results on small/medium-scale benchmarks, including LRA (Tay et al., 2021), ImageNet (Deng et al., 2009), Speech Commands (Warden, 2018), WikiText-103 (Merity et al., 2017) and PG19 (Rae et al., 2019), demonstrate the robust improvements of MEGALODON across scales and modalities.

## 2    Background: Moving Average Equipped Gated Attention (MEGA)

In this section, we setup notations, briefly review the key components in the MEGA architecture (Ma et al., 2023), and discuss the existing problems in MEGA.

Following the notations in MEGA, we use $\boldsymbol{X} = \{\mathbf{x}_1, \mathbf{x}_2, \ldots, \mathbf{x}_n\} \in \mathbb{R}^{n \times d}$ and $\boldsymbol{Y} = \{\mathbf{y}_1, \mathbf{y}_2, \ldots, \mathbf{y}_n\} \in \mathbb{R}^{n \times d}$ to denote the input and output sequences with length $n$, and assume the representations of the input and output sequences have the same dimension $d$.

### 2.1    Multi-dimensional Damped EMA

MEGA embeds an EMA component into the calculation of the attention matrix to incorporate inductive biases across the timestep dimension. Concretely, the multi-dimensional damped EMA first expands each dimension of the input sequence $\boldsymbol{X}$ individually into $h$ dimensions via an expansion matrix $\boldsymbol{\beta} \in \mathbb{R}^{d \times h}$, then applies damped EMA to the $h$-dimensional hidden space. Formally, for each dimension $j \in \{1, 2, \ldots, d\}$:

$$
\begin{aligned}
\mathbf{u}_t^{(j)} &= \boldsymbol{\beta}_j \mathbf{x}_{t,j} \\
\mathbf{h}_t^{(j)} &= \boldsymbol{\alpha}_j \odot \mathbf{u}_t^{(j)} + (1 - \boldsymbol{\alpha}_j \odot \boldsymbol{\delta}_j) \odot \mathbf{h}_{t-1}^{(j)} \\
\mathbf{y}_{t,j} &= \boldsymbol{\eta}_j^T \mathbf{h}_t^{(j)}
\end{aligned}
\tag{1}
$$

where $\mathbf{u}_t^{(j)} \in \mathbb{R}^h$ is the expanded $h$-dimensional vector for the $j$-th dimension at timestep $t$. $\boldsymbol{\alpha} \in (0,1)^{d \times h}, \boldsymbol{\delta} \in (0,1)^{d \times h}$ are the decaying and damping factors, respectively. $\mathbf{h}_t^{(j)} \in \mathbb{R}^h$ is the EMA hidden state for the $j$-th dimension at timestep $t$. $\boldsymbol{\eta} \in \mathbb{R}^{d \times h}$ is the projection matrix to map the $h$-dimensional hidden state back to 1-dimensional output $\mathbf{y}_{t,j} \in \mathbb{R}$.

### 2.2    Moving Average Equipped Gated Attention

In the gated attention mechanism in MEGA, the output from EMA (1) is used to compute the shared representation (Hua et al., 2022), because it encodes contextual information through EMA. Subsequently, MEGA introduces the reset gate, the update gate , and computes the candidate activation with the update gate and the residual connection. The technical details are provided in Appendix A.

### 2.3    Existing Problems in MEGA

To reduce the quadratic complexity in the full attention mechanism, MEGA simply split the sequences of queries, keys and values in (14-16) into chunks of length $c$. The attention in (17) is individually applied to each chunk, yielding linear complexity $O(kc^2) = O(nc)$. Technically, the EMA sub-layer in MEGA helps capture local contextual information near each token, mitigating the problem of losing contextual information beyond chunk boundaries in the chunk-wise attention.

Despite the impressive successes of MEGA, it still suffers its own problems: i) the performance of MEGA with chunk-wise attention still fails behind the one with full attention, due to the limited expressiveness of the EMA sub-layer in MEGA. ii) for different tasks and/or data types, there are architectural divergences in the final MEGA architectures. For example, different normalization layers, normalization patterns (pre-norm vs. post-norm) and attention functions ($f(\cdot)$ in (17)) are applied to different data types (see Ma et al. (2023) for details). iii) There are no empirical evidences showing that MEGA is scalable for large-scale pretraining.

## 3    MEGALODON

To address the aforementioned problems of MEGA, in this section we describe the novel technical advancements of MEGALODON.

### 3.1    CEMA: Extending Multi-dimensional Damped EMA to Complex Domain

As discussed in Ma et al. (2023), the EMA component can be regarded as a simplified state space model with diagonal state matrix. Directly inspired from Gu et al. (2022b), as almost all matrices

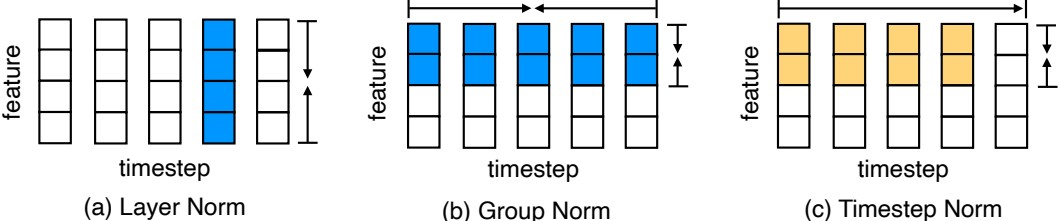

Figure 2: **Normalization methods**. The elements in blue or pink are the regions to compute means and variances. We omit the batch dimension for simplicity.

diagonalize over the complex plane, a straight-forward idea to improve EMA capability is to extend to work over the complex number system $\mathbb{C}$. We propose the *complex exponential moving average (CEMA)*, which re-writes Eq. (1):

$$\mathbf{h}_t^{(j)} = \boldsymbol{\alpha}_j(\cos\theta_j + i\sin\theta_j) \odot \mathbf{u}_t^{(j)} + (1 - \boldsymbol{\alpha}_j \odot \boldsymbol{\delta}_j)(\cos\theta_j + i\sin\theta_j) \odot \mathbf{h}_{t-1}^{(j)}$$
$$\mathbf{y}_{t,j} = \mathrm{Re}(\boldsymbol{\eta}_j^T \mathbf{h}_t^{(j)}) \tag{2}$$

where $\boldsymbol{\alpha}, \boldsymbol{\delta} \in \mathbb{R}^{d \times h}$ are the real number parameters same as in EMA. Different from EMA, $\boldsymbol{\eta} \in \mathbb{C}^{d \times h}$ in CEMA are complex numbers. $\theta_j \in \mathbb{R}^h$, $j \in \{1, 2, \ldots, d\}$ are the $h$ arguments. To uniformly space the $h$ arguments over the period $2\pi$, we parameterize $\theta_j$ as:

$$\theta_{j,k} = \frac{2\pi k}{h}\omega_j, \quad \forall k \in \{1, 2, \ldots, h\} \tag{3}$$

where the learnable parameter $\omega \in \mathbb{R}^d$ depicts the $d$ base angles. By decaying the absolute value of each $h_t$, CEMA preserves the decaying structure in kernel weights, which is a key principle to the success of convolutional models on long sequence modeling (Li et al., 2023c).

### 3.2 Timestep Normalization

Despite the impressive performance of Layer Normalization combined with Transformer, it is obvious that layer normalization cannot directly reduce the internal covariate shift along the spatial dimension (a.k.a timestep or sequential dimension) (Ioffe and Szegedy, 2015). Group Normalization (Wu and He, 2018) normalizes hidden representations both along the timestep dimension and a subset of the feature dimension, which has obtained improvements over Layer Normalization on a range of computer vision tasks. However, it cannot be directly applied to Transformer on auto-regressive sequence modeling, due to the leakage of future information via the mean and variance across the timestep dimension.

In MEGALODON, we extend Group Normalization to the auto-regressive case by computing the cumulative mean and variance. Formally, suppose an input sequence $\boldsymbol{X} = \{\mathbf{x}_1, \mathbf{x}_2, \ldots, \mathbf{x}_n\} \in \mathbb{R}^{n \times d}$, and $k$ groups along the feature dimension with $d_g = d/k$ elements per group. Then, the mean and variance of the first group at timestep $t \in \{1, 2, \ldots, n\}$ are:

$$\mu_t = \frac{1}{t * d_g}\sum_{i=1}^{t}\sum_{j=1}^{d_g} x_{i,j}, \qquad \sigma_t^2 = \frac{1}{t * d_g}\sum_{i=1}^{t}\sum_{j=1}^{d_g}(x_{i,j} - \mu_t)^2 \tag{4}$$

Figure 2 illustrates Layer Normalization and Timestep Normalization. To efficiently and precisely calculate the cumulative mean and variance in each timestep, we provide hardware-friendly implementation on modern hardware (GPU) (see Appendix B.1).

### 3.3 Normalized Attention in MEGALODON

Previous studies have investigated the saturation and instability issues in the original scaled dot-product attention (17). A number of novel techniques have emerged to modify the scaled dot-product attention, among which normalized attention mechanisms, such as (scaled-) cosine attention (Luo et al., 2018; Liu et al., 2022) and QK-normalization (Henry et al., 2020), have stood out for the simplicity and effectiveness.

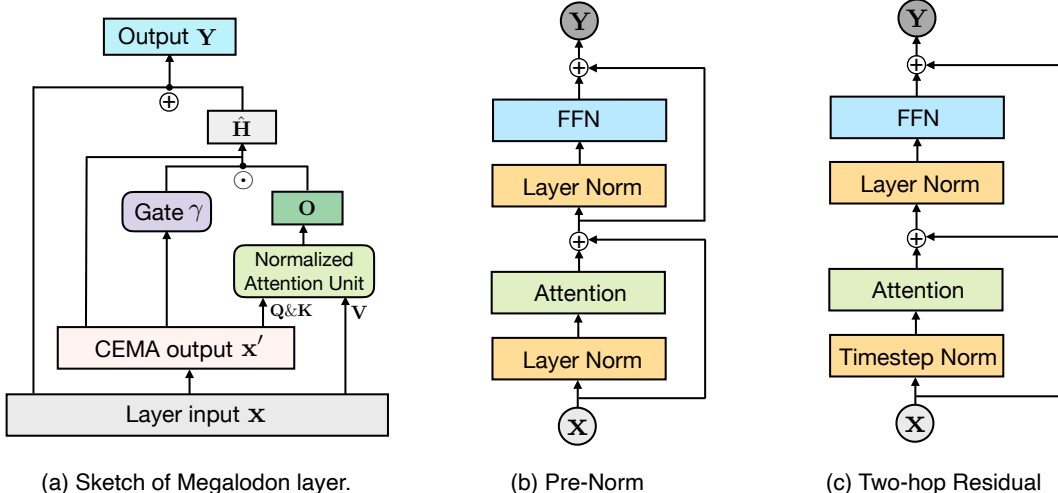

(a) Sketch of Megalodon layer.     (b) Pre-Norm     (c) Two-hop Residual

Figure 3: Illustration of the MEGALODON architecture. Figure (a) shows a sketch of one MEGALODON layer. Figure (b) and (c) display the configurations of pre-norm and pre-norm with two-hop residual, respectively.

Directly inspired from these normalized attention mechanisms, we propose the normalized attention mechanism specifically defined for MEGA to improve its stability. Formally,

$$X' = \text{CEMA}(X) \qquad \in \mathbb{R}^{n \times d} \qquad (5)$$

$$Z = X'W_z + b_z, \quad Z' = \frac{Z}{\|Z\|} \qquad \in \mathbb{R}^{n \times z} \qquad (6)$$

$$Q = \kappa_q \odot Z' + \mu_q \qquad \in \mathbb{R}^{n \times z} \qquad (7)$$

$$K = \kappa_k \odot Z' + \mu_k \qquad \in \mathbb{R}^{n \times z} \qquad (8)$$

where $Q$ and $K$ are computed by using the normalized shared representation $Z'$ instead of $Z$. Note that we remove the SiLU (Ramachandran et al., 2017) activation function $\phi_{\text{silu}}$ in (13), because the normalization on $Z$ has incorporated non-linearity into $Z'$. Then the attention operation in (17) has been changed to:

$$O = f_{\text{softmax}}\left(QK^T\right)V \qquad \in \mathbb{R}^{n \times v} \qquad (9)$$

As we use learnable $\kappa_q$, $\kappa_k$ in (7) and (8), we can remove the scaled term $\tau(X)$. In addition, we found that with the normalized attention, the softmax function $f_{\text{softmax}}$ obtains the best or at least comparable performance on different tasks and data modalities (see Appendix C). Hence, throughout this paper we use softmax as the default attention function.

### 3.4 Pre-Norm with Two-hop Residual

Normalization configurations are crucial in stably training deep architectures, and pre-normalization (Xiong et al., 2020) has become the default normalization configuration because of its better convergence properties than post-normalization in the original Transformer architecture (Vaswani et al., 2017). However, extensive studies have investigated the instability issue of pre-normalization when scaling up model size (Davis et al., 2021; Liu et al., 2022). Formally, a Transformer-based block in pre-noromalization can be formulated as (shown in Figure 3 (b)):

$$\hat{Y} = \text{Attention}(\text{Norm}(X)) + X$$
$$Y = \text{FFN}(\text{Norm}(\hat{Y})) + \hat{Y}$$
$$= \text{FFN}(\text{Norm}(\hat{Y})) + \text{Attention}(\text{Norm}(X)) + X \qquad (10)$$

where the output $Y$ is the sum of the input $X$ and the output of each component in one block. Hence, the range and/or variance of $Y$ keeps increasing for deeper blocks, causing the instability issue. In

the original MEGA architecture, the update gate $\varphi$ (19) is used for a gated residual connection (21) to mitigate this problem (Parisotto et al., 2020; Xu et al., 2020). However, the update gate $\varphi$ introduces more model parameters and the instability issue still exists when scaling up model size to 7 billion.

MEGALODON introduces a new configuration named *pre-norm with two-hop residual*, which simply re-arranges the residual connections in each block (shown in Figure 3 (c)):

$$\hat{Y} = \text{Attention}(\text{Norm}(X)) + X$$
$$Y = \text{FFN}(\text{Norm}(\hat{Y})) + X \tag{11}$$

where the input $X$ is reused as the residual connection of the FFN layer. Since $\hat{Y}$ is directly followed by a normalization layer, we remove the update gate $\varphi$ and use standard residual connection. The graphical architecture of a MEGALODON sub-layer is visualized in Figure 3 (a). Note that the Timestep Normalization is only applied before the attention layer. Before the FFN layer, we still use Layer Normalization. The reasons are two-fold: i) Layer Normalization is faster than Timestep Normalization; ii) the output vector of each token from the attention layer is a mixture of vectors from contextual tokens via attention weights. Hence, normalizing the attention output along the feature dimension is similar to indirectly normalize along the timestep dimension.

### 3.5 4-Dimensional Parallelism in Distributed LLM Pretraining

Efficient distributed training algorithm is essential to train a large-scale language model, and several parallelization mechanisms have been introduced. The three most commonly used parallelism strategies are data, tensor (Shoeybi et al., 2019) and pipeline parallelism (Huang et al., 2019). However, the 3-dimensional parallelism is still insufficient to scale up the context length of LLMs (Li et al., 2023b; Liu et al., 2024).

Benefiting from the chunk-wise attention in MEGALODON, we can efficiently parallelize it along the new timestep/sequence dimension, which is orthogonal to all the aforementioned three parallelism dimensions. In MEGALODON, the only communications between devices in one chunk-parallel group are the last hidden state of CEMA and the cumulative mean and variance of Timestep Normalization in each block. Using asynchronous communication, we can minimize the overhead of chunk parallelization by hiding the communication costs in the computation of other components inside the same block and/or other blocks.

## 4 Experiments

To evaluate the scalability and efficiency of MEGALODON on long-context sequence modeling, we scale up MEGALODON to 7-billion model size and apply it to large-scale language model pretraining on 2 trillion tokens. We also conduct experiments on small/medium-scale sequence modeling benchmarks, including Long Range Arena (LRA) (Tay et al., 2021), raw speech classification on Speech Commands (Warden, 2018), image classification on ImageNet-1K (Deng et al., 2009), and language-modeling on WikiText-103 (Merity et al., 2017) and PG19 (Rae et al., 2019). [2] Empirically, MEGALODON significantly outperforms all the state-of-the-art baseline models on these tasks across various data modalities.

### 4.1 LLM Pretraining

**Architectural Details** In our MEGALODON-7B model, we adopt most of architectural hyperparameters from LLAMA2-7B to ensure fair comparison: MEGALODON-7B consists of 32 blocks, with feature dimension $d = 4096$. Following LLAMA2, we use the SwiGLU activation function (Shazeer, 2020) in the feed-forward layer, and rotary positional embedding (RoPE, Su et al. (2021)). We set the attention chunk size $c = 4096$, which is the same as the pretraining context length in LLAMA2. Benefiting from the attention gate ($\gamma$ in (18)), we use a much smaller number of attention heads $h = 4$ in MEGALODON-7B, comparing to $h = 32$ in LLAMA2-7B. In addition, we apply pre-norm with two-hop residual (§3.4), using Timestep Normalization (§3.2) and Layer Normalization (Ba et al., 2016), while LLAMA2 models apply pre-normalization with RMSNorm (Zhang and Sennrich, 2019).

---

[2]Some results are provided in Appendix C, due to space limits.

**Data and Pretraining Details** We use the same mix of publicly available data from LLAMA2, ensuring that the model are trained on exactly the same 2-trillion tokens. We also use the same tokenizer as LLAMA2, whose vocabulary size is 32K.

We trained MEGALODON-7B using the AdamW optimizer (Loshchilov and Hutter, 2019), with $\beta_1 = 0.9$, $\beta_2 = 0.95$, $\epsilon = 1e-8$. The learning rate is $3.5e-4$ and cosine learning rate schedule is applied with warmup of 2500 steps. We use a weight decay of 0.1 and gradient clipping of 1.0, and no dropout is applied during training. The context length in pretraining is 32K (4 attention chunks). The global batch size is 4M tokens, and is distributed on 256 NVIDIA A100 GPUs (16K tokens per A100). We set data parallel size to 128, chunk parallel size to 2 and tensor parallel size to 1.

**Data and Computation Efficiency** We evaluate the efficiency of MEGALODON w.r.t both the data and computation perspectives. For data efficiency, we display the negative log-likelihood (NLL) for MEGALODON-7B, LLAMA2-7B and LLAMA2-13B w.r.t processed tokens during training in Figure 1. MEGALODON-7B obtains significantly better (lower) NLL than LLAMA2-7B under the same amount of training tokens, demonstrating better data efficiency. Moreover, MEGALODON suffers less training spikes than the Transformer-based architecture in LLAMA2. Note that at the first 1/4 of the pretraining process ($< 500B$ tokens), the NLL of MEGALODON-7B is slightly worse than LLAMA2-7B. We found that the main reason is that we increased the base $\theta$ of RoPE from $10,000$ in LLAMA2 to $100,000$ in MEGALODON, which slows down model convergence at the beginning of the pretraining process. At the end, MEGALODON reaches a training loss of 1.70, landing mid-way between LLAMA2-7B (1.75) and LLAMA2-13B (1.67).

For computation efficiency, we conduct experiments of running LLAMA2-7B and MEGALODON-7B using the same amount of computational resources and comparing their training speed under various context lengths. Specifically, we execute each experiment to train a model with global batch size 4M tokens distributed on 256 NVIDIA A100 GPUs (16K tokens per A100) and calculate the word/token per second (WPS) to measure the training speed. Figure 4 illustrates the average WPS per device of LLAMA2-7B and MEGALODON-7B using 4K and 32K context lengths, respectively. For LLAMA2 models, we accelerate the computation of full attention with Flash-Attention V2 (Dao,

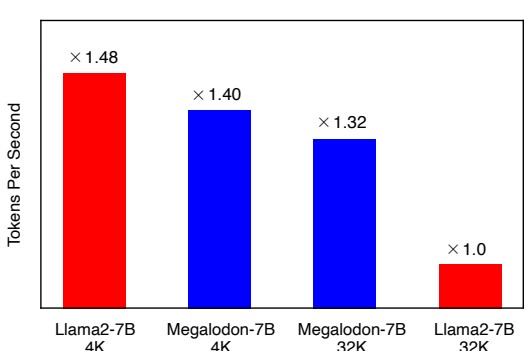

Figure 4: Average WPS per device.

2024). Under 4K context length, MEGALODON-7B is slightly slower (about $6\%$) than LLAMA2-7B, due to the introduction of CEMA and Timestep Normalization. When we scale up context length to 32K, MEGALODON-7B is significantly faster (about $32\%$) than LLAMA2-7B, demonstrating the computation efficiency of MEGALODON for long-context pretraining. In addition, MEGALODON-7B-32K, which utilizes chunk parallelism (§3.5), achieves about $94\%$ utilization of MEGALODON-7B-4K.

## 4.2 Short-Context Evaluation on Academic Benchmarks

We compare MEGALODON-7B to LLAMA2 models on standard academic benchmarks with short contexts ($< 4K$ tokens), closely following the settings in LLAMA2 (Touvron et al., 2023). The benchmarks are grouped into the categories listed below:

- **Commonsense Reasoning** (0-shot): HellaSwag (Zellers et al., 2019), PIQA (Bisk et al., 2020), SIQA (Sap et al., 2019), WinoGrande (Sakaguchi et al., 2021), ARC-e and -c (Clark et al., 2018).

- **World Knowledge** (5-shot): NaturalQuestions (NQ, Kwiatkowski et al. (2019)) and TriviaQA (TQA, Joshi et al. (2017)).

- **Reading Comprehension** (0-shot): BoolQ (Clark et al., 2019).

- **Popular aggregated results** (5-shot): MMLU (Hendrycks et al., 2020).

Table 1 summarizes the results of MEGALODON and LLAMA2 on these academic benchmarks, together with other open-source base models, including MPT (MosaicML, 2023), RWKV (Peng et al., 2023), Mamba (Gu and Dao, 2023), Mistral (Jiang et al., 2023) and Gemma (Mesnard et al., 2024). Pretrained on the same 2T tokens, MEGALODON-7B surpasses LLAMA2-7B across all the

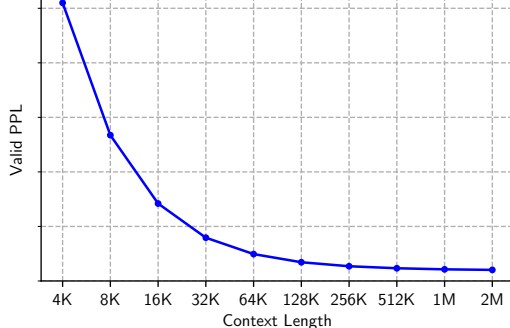

Figure 5: PPL in various context lengths.

Table 2: **Results on Scrolls**. * LLAMA2-L (Xiong et al., 2023) continually trains LLAMA2 on 500B tokens for length extension.

| Model | NaQA | Qasper | QMSum |
|---|---|---|---|
| Xgen | 17.4 | 20.5 | 6.8 |
| MPT | 18.8 | 24.7 | 8.8 |
| Yarn | 20.9 | 26.2 | 11.4 |
| LLAMA2 | 18.8 | 19.8 | 10.1 |
| LLAMA2-L* | 23.5 | **28.3** | **14.5** |
| MEGALODON | **23.9** | 28.0 | 13.1 |

benchmarks. On some tasks, MEGALODON-7B achieves comparable or even better performance with LLAMA2-13B. Note that Mistral-7B and Gemma-8B were pretrained on much larger datasets than MEGALODON-7B, hence the results are not directly comparable.

### 4.3 Long-Context Evaluation

**Perplexity over Long Sequences**   To demonstrate the capability of MEGALODON to make use of very long contexts to improve next-token prediction, we start by conducting the evaluation of valid perplexity on different context lengths. Concretely, we construct a validation dataset which consists of 1,920 selected books. Each of these books contains sequences with at least 2M tokens. The validation dataset is constructed by first randomly shuffling all the files and then concatenating them. Figure 5 shows the perplexity (PPL) of the validation dataset in various context lengths ranging from 4K to 2M. We observe that the PPL decreases monotonically with context length, validating the effectivenss and robustness of MEGALODON on modeling extremely long sequences.

**Long-Context QA tasks in Scrolls**   Next, we evaluate MEGALODON on long-context open-book question answering (QA) tasks in the Scrolls dataset (Shaham et al., 2022), including NarrativeQA (Kočiský et al., 2018), Qasper (Dasigi et al., 2021) and QMSum (Zhong et al., 2021). Following Xiong et al. (2023), we use a simple prompt {CONTEXT} Q: {QUESTION} A: for all the tasks, and evaluate 0-shot F1-score on NarrativeQA, 2-shot F1-score on Qasper and 1-shot geometric-ROUGE[3] on QMSum. Table 2 lists the results of MEGALODON-7B, together with other open-source long-context models in the scale of 7B, namely Xgen-7B-8K (Nijkamp et al., 2023), MPT-7B-8K (MosaicML, 2023), YaRN-7B-128k (Peng et al., 2024), LLAMA2-7B-4K (Touvron et al., 2023) and LLAMA2-7B-32K (LLAMA2-L, Xiong et al. (2023)). MEGALODON-7B obtains the best F1 on NarrativeQA, and competitive results with LLAMA2-7B Long. It should be noticed that LLAMA2-7B Long extends the context length of LLAMA2-7B from 4K to 32K by continually pretraining it on additional 500B tokens from long-context data.

### 4.4 Instruction Finetuning

To evaluation the generalization capability of MEGALODON on instruction following and alignment, We finetune the base model of MEGALODON-7B on a proprietary instruction-alignment data under a controlled setting. We did not apply any RLHF techniques to further finetune it. Table 3 summarizes the performance of chat models in 7B scale on MT-Bench[4]. MEGALODON exhibits superior performance on MT-Bench compared to Vicuna (Chiang et al., 2023), and comparable performance to LLAMA2-Chat, which utilizes RLHF for further alignment finetuning. We present some outputs from instruction finetuned MEGALODON in Appendix D.

Table 3: **MT Bench**. Comparison of Chat models. * LLAMA2-Chat utilizes RLHF.

| Model | Size | MT-Bench |
|---|---|---|
| Vicuna | 7B | 6.17 |
| LLAMA2-Chat* | 7B | 6.27 |
| Mistral-Instruct | 7B | 6.84 |
| MEGALODON | 7B | 6.27 |

---

[3]Geometric mean of ROUGE-1, 2 and L.
[4]https://klu.ai/glossary/mt-bench-eval

Table 4: (**ImageNet-1K**) Top-1 accuracy.

| Model | #Param. | Acc. |
|---|---|---|
| ResNet-152 | 60M | 78.3 |
| ViT-B | 86M | 77.9 |
| DeiT-B | 86M | 81.8 |
| MEGA | 90M | 82.3 |
| MEGALODON | 90M | **83.1** |

Table 5: (**PG-19**) Word-level perplexity.

| Model | #Param. | Val | Test |
|---|---|---|---|
| Compressive Trans. | – | 43.4 | 33.6 |
| Perceiver AR | 975M | 45.9 | 28.9 |
| Block-Recurrent Trans. | 1.3B | – | 26.5 |
| MEGABYTE | 1.3B | 42.8 | 36.4 |
| MEGALODON | 1.3B | **29.5** | **25.4** |

## 4.5 Evaluation on Medium-Scale Benchmarks

**ImageNet Classification**   To evaluate MEGALODON on image classification task, we conduct experiments on the Imagenet-1K (Deng et al., 2009) dataset, which consists of 1.28M training images and 50K validation images from 1000 classes. We mostly follow DeiT's approach of applying several data augmentation and regularization methods that facilitate the training process, and adopt most the hyperparameters from Ma et al. (2023). For classification task, we replace the timestep normalization with the standard group normalization method. Top-1 accuracy on the validation set is reported in Table 4 to assess various models. MEGALODON obtains about 1.3% accuracy improvement over DeiT-B (Touvron et al., 2021), and 0.8%. improvement over MEGA (Ma et al., 2023).

**Auto-regressive Language Modeling on PG-19**   We also evaluate MEGALODON on auto-regressive language modeling on the medium-scale PG19 (Rae et al., 2019) datasets. We use the same vocabulary from Block-Recurrent Transformer (Hutchins et al., 2022) and adopt most of its hyper-parameters to train a MEGALODON model with 1.3B parameters. Table 5 illustrate the word-level perplexity (PPL) of MEGALODON on PG-19, together with previous state-of-the-art models, including Compressive Transformer (Rae et al., 2020), Perceiver AR (Hawthorne et al., 2022), Block-Recurrent Transformer (Hutchins et al., 2022) and MEGABYTE (Yu et al., 2024). MEGALODON significantly outperforms all the baselines.

## 5   Conclusion

We have introduced MEGALODON, an improved MEGA architecture with multiple novel technical components, including complex exponential moving average (CEMA), the timestep normalization layer, normalized attention and pre-norm with two-hop residual configuration, to improve its capability, efficiency and scalability. Through a direct comparison with LLAMA2, MEGALODON achieves impressive improvements on both training perplexity and across downstream benchmarks. Importantly, experimental results on long-context modeling demonstrate MEGALODON's ability to model sequences of unlimited length. Additional experiments on small/medium-scale benchmarks across different data modalities illustrate the robust improvements of MEGALODON, which lead to a potential direction of future work to apply MEGALODON for large-scale multi-modality pretraining.

## Acknowledgments

We thank Sadhika Malladi, Zihao Ye, Dacheng Li and Rulin Shao for their helpful feedback and discussion during this work.

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

# Appendix: MEGALODON: Efficient Long-Context LLM Pretraining and Inference with Unlimited Context Length

## A Background: Moving Average Equipped Gated Attention

In the gated attention mechanism in MEGA, the output from EMA (1) is used to compute the shared representation (Hua et al., 2022) $\boldsymbol{Z}$:

$$\boldsymbol{X}' = \text{EMA}(\boldsymbol{X}) \qquad\qquad \in \mathbb{R}^{n \times d} \qquad (12)$$

$$\boldsymbol{Z} = \phi_{\text{silu}}(\boldsymbol{X}'W_z + b_z) \qquad\qquad \in \mathbb{R}^{n \times z} \qquad (13)$$

where $\boldsymbol{X}'$ can be regarded as the updated or contextual input, because it encodes contextual information through EMA. Then, the query and key sequences are computed by applying per-dimension scalars and offsets to $\boldsymbol{Z}$, and the value sequence is from the original $\boldsymbol{X}$:

$$\boldsymbol{Q} = \boldsymbol{\kappa}_q \odot \boldsymbol{Z} + \boldsymbol{\mu}_q \qquad\qquad \in \mathbb{R}^{n \times z} \qquad (14)$$

$$\boldsymbol{K} = \boldsymbol{\kappa}_k \odot \boldsymbol{Z} + \boldsymbol{\mu}_k \qquad\qquad \in \mathbb{R}^{n \times z} \qquad (15)$$

$$\boldsymbol{V} = \phi_{\text{silu}}(\boldsymbol{X}W_v + b_v) \qquad\qquad \in \mathbb{R}^{n \times v} \qquad (16)$$

where $\boldsymbol{\kappa}_q, \boldsymbol{\mu}_q, \boldsymbol{\kappa}_k, \boldsymbol{\mu}_k \in \mathbb{R}^z$ are the learnable scalars and offsets of queries and keys, respectively. $v$ is the expanded intermediate dimension for the value sequence. The output of attention is computed as follows:

$$\boldsymbol{O} = f\left(\frac{\boldsymbol{Q}\boldsymbol{K}^T}{\tau(\boldsymbol{X})}\right)\boldsymbol{V} \qquad\qquad \in \mathbb{R}^{n \times v} \qquad (17)$$

Subsequently, MEGA introduces the reset gate $\boldsymbol{\gamma}$, the update gate $\boldsymbol{\varphi}$, and computes the candidate activation $\hat{\boldsymbol{H}}$ and final output $\boldsymbol{Y}$:

$$\boldsymbol{\gamma} = \phi_{\text{silu}}(\boldsymbol{X}'W_\gamma + b_\gamma) \qquad\qquad \in \mathbb{R}^{n \times v} \qquad (18)$$

$$\boldsymbol{\varphi} = \phi_{\text{sigmoid}}(\boldsymbol{X}'W_\varphi + b_\varphi) \qquad\qquad \in \mathbb{R}^{n \times d} \qquad (19)$$

$$\hat{\boldsymbol{H}} = \phi_{\text{silu}}(\boldsymbol{X}'W_h + (\boldsymbol{\gamma} \odot \boldsymbol{O})U_h + b_h) \qquad\qquad \in \mathbb{R}^{n \times d} \qquad (20)$$

$$\boldsymbol{Y} = \boldsymbol{\varphi} \odot \hat{\boldsymbol{H}} + (1 - \boldsymbol{\varphi}) \odot \boldsymbol{X} \qquad\qquad \in \mathbb{R}^{n \times d} \qquad (21)$$

with the update gate $\boldsymbol{\varphi}$ and the residual connection $\boldsymbol{X}$.

## B Implementation Details

### B.1 Efficient Fused CUDA Operators Implementation

**Fused Attention** We implemented a fused attention operator to improve the efficiency, mainly by fusing the causal mask, softmax function and dropout operation (if necessary). The fused implementation reduces the IO costs from global memory for the attention weight. For attention dropout, we adopt the dropout-before-softmax scheme in DropKey (Li et al., 2023a), which applies the dropout mask on the input attention matrix of the softmax function. Concretely, we fill the values of the attention matrix at dropout mask positions to $-\infty$ before feeding it into the softmax function. One important advantage of this dropout-before-softmax scheme comparing to the standard attention dropout is that the computation of the gradients in back-propagation is independent with the applied dropout mask.

**Efficient FFTConv** We also provide an efficient fused implementation of the FFTConv operator. Similar with the FlashConv in H3 (Fu et al., 2023), we fused the real number FFT (RFFT), its inverse (IRFFT) and implemented the Cooley-Tukey FFT algorithm (Cooley and Tukey, 1965) in the CUDA shared memory. Similar with the FlashConv in H3 (Fu et al., 2023), we fused the real number FFT (RFFT), its inverse (IRFFT) and the element-wise multiplication, and implemented the Cooley-Tukey FFT algorithm (Cooley and Tukey, 1965) in CUDA's shared memory. Our implementation is able to accommodate up to 16K tokens in the limited shared memory of an A100 GPU.

Table 6: (**Long Range Arena**) Accuracy on the full suite of long range arena (LRA) tasks. Results of previous models are reported in Ma et al. (2023).

| Models | ListOps | Text | Retrieval | Image | Pathfinder | Path-X | Avg. |
|---|---|---|---|---|---|---|---|
| Transformer | 37.11 | 65.21 | 79.14 | 42.94 | 71.83 | ✗ | 59.24 |
| Reformer | 37.27 | 56.10 | 53.40 | 38.07 | 68.50 | ✗ | 50.67 |
| Linformer | 35.70 | 53.94 | 52.27 | 38.56 | 76.34 | ✗ | 51.36 |
| BigBird | 36.05 | 64.02 | 59.29 | 40.83 | 74.87 | ✗ | 55.01 |
| Luna-256 | 37.98 | 65.78 | 79.56 | 47.86 | 78.55 | ✗ | 61.95 |
| S4 | 59.10 | 86.53 | 90.94 | 88.48 | 94.01 | 96.07 | 85.86 |
| MEGA-chunk | 58.76 | 90.19 | 90.97 | 85.80 | 94.41 | 93.81 | 85.66 |
| MEGA | 63.14 | 90.43 | 91.25 | **90.44** | 96.01 | 97.98 | 88.21 |
| MEGALODON-chunk | 62.23 | **90.53** | 91.74 | 87.11 | 96.89 | 97.21 | 87.62 |
| MEGALODON | **63.79** | 90.48 | **91.76** | 89.42 | **98.13** | **98.17** | **88.63** |

**Timestep Normalization**    For the TimestepNorm operator, we have an efficient implementation to improve both its speed and numerical stability. To compute the cumulative mean and variance for each of the timesteps, our implementation distributed the threads in each CUDA block in both the timestep/sequence dimension and the feature dimension to balance the parallelism of the algorithm and the performance of the global memory access. To improve numerical stability, we used the Welford algorithm (Welford, 1962) to compute the cumulative mean and variance and the Kahan Summation (Kahan, 1965) to reduce the numerical error from summation.

### B.2   Plus 1 Reparameterization in Normalization Layers

In the normalization methods, two learnable parameters $\gamma$ and $\beta$ are introduced to scale and shift the normalized value:

$$y = \gamma \frac{x - \mu}{\sigma} + \beta \tag{22}$$

where $\mu$ and $\sigma^2$ are the mean and variance of the input $x$ across the pre-defined dimensions. Initialization of $\gamma$ and $\beta$ is crucial for model performance and stability. The standard implementation of normalization layers, such as PyTorch (Paszke et al., 2019), initializes $\gamma$ and $\beta$ to vectors of ones and zeros, respectively, to preserve the mean and variance of the normalized inputs at the beginning of training.

This standard implementation, however, suffers a problem when weight decay regularization is applied to prevent overfitting (Hanson and Pratt, 1988). Technically, the weight decay regularization pushes the values of model parameters towards smaller magnitudes. In the context of normalization methods, weight decay pushes the values in $\gamma$ towards zero, which diverges from its initialization of one. This may prevent the model from learning the true scale of the data distribution, and may cause numerical stability issues as well.

To address this problem, we used the *plus 1 reparameterization*[5] of the scale parameter $\gamma$:

$$y = (\gamma + 1) \frac{x - \mu}{\sigma} + \beta \tag{23}$$

where $\gamma$ is initialized to zero. Under weight decay, $\gamma$ remains centered around zero, resulting in a desirable scale of $\gamma + 1$ around one.

## C   Experiments on Small-Scale Benchmarks

We conducted small-scale experiments on five benchmarks across various data modalities, including text, audio and image. To demonstrate the robustness of the MEGALODON architecture on different tasks and data types, we used a single unified architecture with minimal architectural divergence in

---

[5]Similar idea in the blog: https://medium.com/@ohadrubin/exploring-weight-decay-in-layer-normalization-challenges-and-a-reparameterization-solution-ad4d12c24950

| Table 7: (**SC-Raw**) Accuracy. | | |
| --- | --- | --- |
| **Model** | **#Param.** | **Acc.** |
| Transformer | 786K | ✗ |
| S4 | 300K | 97.50 |
| MEGA | 300K | 96.92 |
| MEGA (big) | 476K | 97.30 |
| MEGALODON | 300K | **98.14** |

| Table 8: (**WikiText-103**) Word-level PPL. | | |
| --- | --- | --- |
| **Model** | **#Param.** | **PPL** |
| Transformer | 247M | 18.66 |
| Transformer-XL | 257M | 18.30 |
| S4 | 249M | 20.95 |
| MEGA | 252M | 18.07 |
| MEGALODON | 252M | **17.23** |

all the experiments: softmax attention function, rotary postional embedding, pre-norm with two-hop residual, and timestep Normalization (Group Normalization for classification). We adopt (almost) all the architectural and training hyperparameters from the corresponding experiments of the original MEGA (Ma et al., 2023).

## C.1 Long Range Arena (LRA)

Long Range Arena (LRA) benchmark (Tay et al., 2021) is designed for evaluating sequence models under the long-context scenario. They collect six tasks in this benchmark which are ListOps (Nangia and Bowman, 2018), byte-level text classification (Text; Maas et al. (2011)), byte-level document retrieval (Retrieval; Radev et al. (2013)), image classification on sequences of pixels (Image; Krizhevsky et al. (2009)), Pathfinder (Linsley et al., 2018) and its extreme long version (Path-X; Tay et al. (2021)). These tasks consist of input sequences ranging from 1K to 16K tokens and span across a variety of data types and modalities.

Table 6 compares MEGALODON against several baselines, including Transformer and its efficient variants, the state space model S4 (Gu et al., 2022a), and the original MEGA model. Following Ma et al. (2023), we also evaluate MEGALODON-chunk on each task, by setting the chunk size $c = 128$ for all the tasks, except Path-X where $c = 4096$. With chunk-wise attention, MEGALODON-chunk substantially outperforms MEGA-chunk on all the six tasks. In addition, MEGALODON significantly narrows the gap between chunk-wise attention and full attention.

## C.2 Raw Speech Classification

To evaluate the capability of MEGALODON on the long-range modeling of speech signals, we apply MEGALODON to classify raw speech (with length 16000), rather than using traditional preprocessing (e.g. convert to MFCC features). Following Ma et al. (2023), we perform speech classification on the SC10 subset of the Speech Commands dataset (Warden, 2018), and set attention chunk size $c = 1000$. As shown in Table 7, our MEGALODON model with 300K parameters achieves an accuracy of 98.14, which is significantly better than 97.50 from S4 and 96.92 from MEGA with similar model size.

## C.3 Auto-regressive Language Modeling

We also evaluate MEGALODON on auto-regressive language modeling on the small-scale WikiText-103 (Merity et al., 2017) datasets. Following Baevski and Auli (2018), we adopt adaptive softmax and input embeddings and use a vocabulary of 260K tokens for WikiText-103 and train a MEGALODON model with about 250M parameters with attention chunk size $c = 2048$. Table 8 illustrate the word-level perplexity (PPL) of MEGALODON on WikiText-103, together with previous state-of-the-art models, including Transformer (Baevski and Auli, 2018), Transformer-XL (Dai et al., 2019), S4 (Gu et al., 2022a) and MEGA (Ma et al., 2023). MEGALODON significantly outperforms all the baselines on both the two datasets.

# D Model Outputs from Instruction-finetuned MEGALODON

**What is the central dogma of molecular biology? What processes are involved? Who named this?**

The central dogma of molecular biology describes the flow of genetic information from DNA to RNA to proteins. The three processes involved in the central dogma are transcription, translation, and translocation.
The central dogma was first proposed by Francis Crick, James Watson, and Maurice Wilkins in the 1950s, and was later formalized by Crick in a 1958 paper titled "On the Central Dogma of Molecular Biology". The term "central dogma" was coined by Crick to describe the flow of genetic information from DNA to RNA to proteins.

**Identify and fix one incorrect fact in your previous response.**

The correct name of the person who proposed the central dogma of molecular biology is James Watson, not Maurice Wilkins. James Watson, along with Francis Crick, first proposed the central dogma in the 1950s, and Maurice Wilkins was a co-recipient of the Nobel Prize in Physiology or Medicine in 1962 for his contributions to the discovery of the double helix structure of DNA.

**Act as a math teacher. I will provide some mathematical equations or concepts, and it will be your job to explain them in easy-to-understand terms. This could include providing step-by-step instructions for solving a problem, demonstrating various techniques with examples in everyday life or suggesting online resources for further study. My first request is "I need help understanding how probability works."**

Probability is a branch of mathematics that deals with the likelihood of an event occurring. It is often expressed as a number between 0 and 1, where 0 represents an impossible event and 1 represents a certain event.
For example, if you were to roll a fair six-sided die, the probability of rolling a 1 would be 1/6, or about 17%. This is because there is only one way for the die to land on a 1 out of the six possible outcomes.
To calculate probability, you can use the formula:
Probability = Number of ways the event can occur / Total number of possible outcomes
For example, if you were to ask "What is the probability of flipping heads on a fair coin toss?", the answer would be 1/2, or 50%. This is because there is only one way for the coin to land heads-up out of the two possible outcomes (heads or tails).
There are many different ways to interpret probability, and it can be used in a variety of real-world applications such as gambling, insurance, and statistics.
I hope this helps to give you a basic understanding of probability. Let me know if you have any questions or need further clarification.

**What are the differences between Riemannian geometry and euclidean geometry?**

Riemannian geometry and Euclidean geometry are both branches of mathematics that study the properties of space, but they differ in their assumptions and methods.
Euclidean geometry is based on the assumption that the space is flat and flat Euclidean space is the simplest non-empty connected flat metric space. It studies the properties of space using concepts such as distance, angle, and parallelism. It is named after the Greek mathematician Euclid, who developed the foundations of the field in his book "Elements".
Riemannian geometry, on the other hand, is based on the assumption that the space is curved. It studies the properties of space using concepts such as distance, angle, and parallelism, but in a more general setting, and it is named after the German mathematician Bernhard Riemann.
In Riemannian geometry, the concept of a metric is used to measure the distance between two points in a curved space. A metric is a mathematical function that assigns a distance to each segment in a space. The most common metric in Riemannian geometry is the distance function induced by a Riemannian metric, which is a mathematical function that assigns a distance to each segment in a space.
In summary, Euclidean geometry is a special case of Riemannian geometry, where the metric is the flat Euclidean metric. Riemannian geometry is a more general field that studies the properties of curved spaces, and it is used in many areas of mathematics, physics and other sciences.
I hope this helps to give you a basic understanding of the differences between Riemannian geometry and Euclidean geometry. Let me know if you have any questions or need further clarification.

**Evaluate the following movie reviews on a scale of 1 to 5, with 1 being very negative, 3 being neutral, and 5 being very positive: 1. This movie released on Nov. 18, 2019, was phenomenal. The cinematography, the acting, the plot - everything was top-notch. 2. Never before have I been so disappointed with a movie. The plot was predictable and the characters were one-dimensional. In my opinion, this movie is the worst one to have been released in 2022. 3. The movie was okay. There were some parts I enjoyed, but there were also parts that felt lackluster. This is a movie that was released in Feb 2018 and seems to be quite ordinary. Return the answer as a JSON array of integers.**

[5, 1, 3]

**Update your previous reply by including the release date as part of the JSON content.**

```
[
    {"release_date": "Nov 18, 2019",    "rating": 5},
    {"release_date": "2022", "rating": 1},
    {"release_date": "Feb 2018", "rating": 3}
]
```

Figure 6: Model outputs from instruction-finetuned MEGALODON.

