# OpenReview forum: "Megalodon: Efficient LLM Pretraining and Inference with Unlimited Context Length"
_NeurIPS.cc/2024/Conference — NeurIPS 2024 poster_

### Official Review · Reviewer_xs5D · 2024-07-10

**Soundness:** 3
**Presentation:** 3
**Contribution:** 2
**Rating:** 5
**Confidence:** 3

**Summary:**

The paper introduces Megaladon, an improvement on the existing technique Mega. This technique uses a diagonalizable complex moving average to allow for integration of information across a longer context efficiently.

**Strengths:**

- The technique is validated against modern models across a large variety of benchmarks
- The diagonalizable complex moving average seems like an easy win for efficient computation and improvement
- The scale of the models produced is quite large and provides clear signal, while also producing useful model artifacts

**Weaknesses:**

The evaluation is lacking in some key aspects. If these are remedied, the paper is strong.

- There needs to be more comparison against MEGA, this seems critical to establish technical novelty beyond operating at a larger scale than the Mega paper.
At scale, for the core NLP benchmarks, how does Megaladon compare Mega and Mega chunk? What changes actually make an improvement? Ablations in particualar around these design choices seem key to establishing the technical novelty, since the changes are somewhat small and it's not clear they buy you a large gap in improvement, or which pieces do.

I have looked at the Imagenet Mega comparison, and those in the appendix, but this is not at scale, which is a critical element of the paper.

- The paper spends much time comparing against models of shorter context lengths, while it is clear the unlimited inference context length is an advantage, Mega had this too. A compute matched setting should still be compared against. In particular, How does Llama trained with let's say a ~28k context (or whatever context yields the same 1.3x speed improvement of Megaladon) perform in comparison at the scale of the main experiments on the main benchmarks.  The monotonic perplexity in Figure 5 is nice to see, but the perplexity other methods at their native or extended context lengths should be included, the minimum perplexity by megaladon might be higher than those achieved by other methods at shorter context lengths.

- A commonly targeted use case of models like this is direct byte level modeling, it would be interesting to see how this method performs on that task. Many of the methods in the long context section were designed with this purpose explicitly in mind. This seems like an application which is important to test on, particularly if context extension is viable in this domain using this method, that would be very interesting.

- Mamba state space models seem to exhibit naturally good context extension, since this method can be seen as a state space model it might be good to compare in the long sequence domain at a similar scale to the 3b Mamba-[1,2] models.


While it might be infeasible to include all of the above, the first 2-3 seem of particular importance to establish the usefulness and technical novelty of this technique.

**Questions:**

Why do not start to modeling improvements until well over 1T training tokens, is this a byproduct of the learning schedule or of something structural within the model?

How much compute is added over Mega/Mega-Chunk to incorporate the necessary changes?

How is the complex moving average implemented?

**Limitations:**

Limitations should be addressed further and more directly in the paper.

---

> ### Author Rebuttal · Authors · 2024-08-06
>
> Thanks for your time and constructive comments! We appreciate your positive feedback on the good motivation, novelty of Megalodon and its strong empirical results. We address your concerns and questions below and please let us know if you still have concerns after you read our response.
>
> > W1: There needs to be more comparison against MEGA, this seems critical to establish technical novelty beyond operating at a larger scale than the Mega paper…
>
> The ablation studies on small/mederatel-scale benchmarks are in the [author rebuttal](https://openreview.net/forum?id=XlAbMZu4Bo&noteId=G72sD1BBxi).
>
> We cannot successfully train Mega on a larger-scale setting, due to numerical instability. That is why we did not compare Megalodon with Mega with large scale experiments.
>
> > W2:  In particular, How does Llama trained with let's say a ~28k context (or whatever context yields the same 1.3x speed improvement of Megaladon) perform in comparison at the scale of the main experiments on the main benchmarks.
>
> The training flops of Megalodon-7B is similar with LlaMa2-7B with 4K context, because we also used 4K as the chunk size of chunk-wise attention. To keep the same 1.3x speed over Transformer with 32K context length, the context length of full attention should be at most 8K-10K.
>
> > W3: The monotonic perplexity in Figure 5 is nice to see, but the perplexity other methods at their native or extended context lengths should be included, the minimum perplexity by megaladon might be higher than those achieved by other methods at shorter context lengths.
>
> [Previous studies](https://github.com/jzhang38/LongMamba) have investigated the PPL of Transformer and other architectures (such as Mamba) beyond their training context lengths. Without any fine-tuning for length extension, Transformer and Mamba did not achieve the monotonic PPL of Megalodon in Figure 5.
>
> > W4: A commonly targeted use case of models like this is direct byte level modeling, it would be interesting to see how this method performs on that task.
>
> We appreciate your suggestion on evaluating Megalodon on byte-level modeling tasks. In our experiments on PG-19 (Table 5), we compared Megalodon with MegaByte, which is one of the most advanced byte-level architectures. In addition, among the six tasks in LRA (Table 6), the three text-classification tasks are in byte level. These results indirectly demonstrate the effectiveness of Megalodon on byte-level modeling. We leave more advanced byte-level modeling with Megalodon to future work.
>
> > W5: Mamba state space models seem to exhibit naturally good context extension, since this method can be seen as a state space model it might be good to compare in the long sequence domain at a similar scale to the 3b Mamba-[1,2] models.
>
> [Previous study](https://github.com/jzhang38/LongMamba) showed that, without any fine-tuning,  the context extension of Mamba is not as good as Megalodon.
>
> > Q1: Why do not start to modeling improvements until well over 1T training tokens, is this a byproduct of the learning schedule or of something structural within the model?
>
> There are different factors to impact the convergence speed at the beginning of training, including peak learning rate, learning rate scheduler and warmup steps. Thus we believe the final pre-trianing loss is more indicative.
>
> > Q2: How much compute is added over Mega/Mega-Chunk to incorporate the necessary changes?
>
> The additional flops of Megalodon over Mega are from complex CEMA (over real-number EMA) and TImestep Normalization (over Layer/RMS Normalization). These additional flops are marginal (< 1%) compared to the total flops in Megalodon.
>
> > Q3: How is the complex moving average implemented?
>
> The implementation of CEMA is similar to that of EMA. We use FFT to compute the outputs (see appendix A in Mega paper for details). To accelerate CEMA, we implemented separated CUDA kernels for the computation of the convolutional kernel and FFT.

---

> > ### Comment · Reviewer_xs5D · 2024-08-08
> > **Nearly There**
> >
> > Hi,
> > Thank you for the thoughtful response.
> >
> > The ablations help, and knowing that the changes were critical to allowing Mega to achieve larger scales is good. Knowing that there are byte level classification present.
> >
> > For the monotonic perplexity plot, can you please add in a y-axis scale and the the perplexities of the other techniques at their native context length. The moving average makes it clear that generalization is possible, but it's critical to know the range of improvement we can actually expect. The actual improvement may not be that much though.
> >
> > With this final change, I will raise my score.
> >
> > Thank you

---

> > > ### Author Response · Authors · 2024-08-08
> > > **Re: Nearly There**
> > >
> > > We appreciate your positive response!
> > >
> > > > For your suggestion on reporting PPLs of other architectures at their native context length.
> > >
> > > The problem is that the PPLs from different models on the held-out validation dataset are not directly comparable, since these models were pre-trained on different training data. The only comparable model/architecture is LlaMa2-7B because it was trained on exactly the same 2T data with Megalodon. However, LlaMa2 only supports 4K context length. On 4K context length, the PPL of LlaMa2-7B on the validation data in Figure 5 is $5.09$ while the corresponding PPL of Megalodon is $4.98$. Thus, even at a relatively short context length of 4K, Megalodon slightly outperforms LlaMa2 on PPL.

---

> > > > ### Comment · Reviewer_xs5D · 2024-08-09
> > > > **Re: Nearly There**
> > > >
> > > > I do still think the perplexities should be comparable if we expect the downstream performance numbers to be comparable, we should also expect the perplexities to be comparable as well correct?
> > > >
> > > > At the very least the y-xis should be labeled on that plot, is the perplexity monotonically decreasing from 4.98 to 4.97 or down much further for example. It is unclear.

---

> > > > > ### Author Response · Authors · 2024-08-09
> > > > > **Re: Nearly There**
> > > > >
> > > > > > I do still think the perplexities should be comparable if we expect the downstream performance numbers to be comparable, we should also expect the perplexities to be comparable as well correct?
> > > > >
> > > > > Unfortunately, neither the PPL nor the downstream performance are directly comparable when the models are in different scales (#parameters) and/or trained on different datasets. That is why we trained our Megalodon-7B on exactly the same 2T training tokens with LlaMa2-7B. Therefor, the only comparable downstream results in Table 1 are LlaMa2-7B and Megalodon-7B. And that is why we put them together in the last two rows. Conducting a direct comparison between Megalodon and the Transformer architecture in LlaMa2 in large-scale pretraining is one of the main motivations and contributions of this work.
> > > > >
> > > > > > At the very least the y-xis should be labeled on that plot, is the perplexity monotonically decreasing from 4.98 to 4.97 or down much further for example. It is unclear.
> > > > >
> > > > > As mentioned in Section 4.3, the validation data in Figure 5 are selected books, each of which contains at least 2M tokens. We removed the scale in y-xis in Fiture 5 to protect the privacy of the data. What we can say is that the PPL at 4K is $4.98$ and the PPL at 2M is $4.82$

---

### Official Review · Reviewer_6wcE · 2024-07-13

**Soundness:** 3
**Presentation:** 3
**Contribution:** 3
**Rating:** 5
**Confidence:** 5

**Summary:**

Megalodon i.e. Mega2 improves over Mega by using (1) complex-valued EMA; (2) improved normalization schemes (e.g. timestep normalization, attention normalization, pre-norm with 2-hop residuals).

**Strengths:**

- Timestep normalization is simple and reasonable. A highly-optimized CUDA kernel provided in this work could be very influential - it is possible to be as popular as layernorm/groupnorm in the future.

- Gated attention is verified in large-scale training setting for the first time.

**Weaknesses:**

- There are no ablation studies in moderate-scale language modeling at all

- Figure1: the training sequence length is not the same, thus the perplexity comparison might be unfair.

- Lacking discussions & comparison to recent hybrid (local) attention and RNN models: e.g. Griffin, Jamba

- Long context evaluation is not comprehensive: the paper claims unlimited context length, while it is unclear about the effective context length. It is better to provide the Needle in the Haystack results

**Questions:**

- Complex-valued linear recurrence is replaced by gated linear recurrence for many recent works:  e.g. LRU (complex) -> RG-LRU (real gated). HGRN1 (complex) -> HGRN2 (real gated).  S4 (complex) -> Mamba1 & 2 (real gated).  Did you try real-valued gated linear recurrence layers?

- What's your opinion on Sliding Window Attention vs. Chunk Attention?

---

> ### Author Rebuttal · Authors · 2024-08-06
>
> Thanks for your time and constructive comments! We appreciate your positive feedback on the good motivation, novelty of Megalodon and its strong empirical results. We address your concerns and questions below and please let us know if you still have concerns after you read our response.
>
> > W1: There are no ablation studies in moderate-scale language modeling at all
>
> The ablation studies on small/mederate-scale benchmarks are in the [author rebuttal](https://openreview.net/forum?id=XlAbMZu4Bo&noteId=G72sD1BBxi).
>
> > W2: Figure1: the training sequence length is not the same, thus the perplexity comparison might be unfair.
>
> Though the training sequence length is not the same, the Megalodon-7B model is using 4K attention chunk size, which is the same as the context length of full attention in LlaMa2-7B. Thus Megalodon-7B and LlaMa2-7B in Figure 1 were trained on the same data with similar flops/token, yielding relatively fair PPL comparison.
>
> > W3: Lacking discussions & comparison to recent hybrid (local) attention and RNN models: e.g. Griffin, Jamba
>
> Griffin and Jamba are two concurrent works and they are trained in different scales of model size and data.
>
> > W4: Long context evaluation is not comprehensive: the paper claims unlimited context length, while it is unclear about the effective context length. It is better to provide the Needle in the Haystack results
>
> We conducted experiments to evaluate Megalodon on retrieval-oriented tasks, such as passkey retrieval. Similar to previous studies, due to chunk-wise attention, Megalodon under-performed on these retrieval-oriented tasks compared with full attention mechanism, but outperformed state-space models such as Mamba.
>
> For example, without any fine-tuning for length extension, Megalodon completes the passkey retrieval task with up to 16K context length, while Mamba can only complete this task up to 4K context. Long-LlaMa2, which continually trains LlaMa2 on selected long-context data for length extension, is able to complete up to 32K context length.
>
> Further improving Megalodon for retrieval-oriented tasks is an interesting and important direction for future work.
>
> > Q1: Complex-valued linear recurrence is replaced by gated linear recurrence for many recent works: e.g. LRU (complex) -> RG-LRU (real gated). HGRN1 (complex) -> HGRN2 (real gated). S4 (complex) -> Mamba1 & 2 (real gated). Did you try real-valued gated linear recurrence layers?
>
> Gated linear recurrence, such as Mamba, HGRN1 etc, incorporates input-dependent recurrence, which sacrifices efficiency compared to (complex) EMA and S4. We did not try gated linear recurrence mainly because Megalodon leverages chunk-wise attention mechanism, which might down-weight the necessity of input-dependent recurrence.
>
> > Q2: What's your opinion on Sliding Window Attention vs. Chunk Attention?
>
> Sliding Window Attention is theoretically more powerful but less efficient than Chunk-wise Attention. Moreover, sliding window attention increases the difficulty of parallel training along the chunk/context parallel groups. It is an interesting direction for future work to efficiently incorporate sliding window attention into Megalodon.

---

> > ### Comment · Reviewer_6wcE · 2024-08-08
> >
> > Regarding ablations, it is well known that performance on wikitext-103 can be sensitive to regularization since the dataset is relatively small, can we have some experiments on Slimpajama?

---

> > > ### Author Response · Authors · 2024-08-08
> > > **Re: Official Comment by Reviewer 6wcE**
> > >
> > > We appreciate your suggestion on conducting ablation studies on Slimpajama. However, due to the limits of time and computation resources, we cannot provide the results during rebuttal period. We will consider to add this study in our final version.
> > >
> > > In addition, though the performance on WikiText-103 might be impacted by regularization, we want to say that we have fully swept hyper-parameters for the models in our ablation study to ensure fair comparison. The results in our table are better than previous state-of-the-art numbers, demonstrating the reliability of our ablation study.

---

> > > > ### Comment · Reviewer_6wcE · 2024-08-13
> > > >
> > > > I appreciate the large-scale training done in this work and will maintain my positive score, leaning towards acceptance. Here are some suggestions for the camera-ready version:
> > > >
> > > > - I understand that it may be challenging to incorporate many results during the rebuttal period. However, I highly recommend that the authors conduct some moderate-scale experiments for ablation in the camera-ready version, which I believe would significantly benefit the paper. I suggest using the setting from Samba [1], specifically a 1.3B model trained on Slimpajama with 100Bn tokens.
> > > >
> > > > - I would also suggest revising the claim of "unlimited context length" in the camera-ready version. This is definitely an overstatement. While I understand that many works, such as [1], make similar claims, from a more scientific perspective, the ability to process arbitrary sequence lengths differs from effectively utilizing the entire context. I can easily imagine that these RNN + local attention models still struggle with retrieval tasks like Needle in the Haystack or Phonebook Lookup. Please include related discussions in the camera-ready version.
> > > >
> > > > [1] Samba: Simple Hybrid State Space Models for Efficient Unlimited Context Language Modeling

---

### Official Review · Reviewer_73L7 · 2024-07-14

**Soundness:** 2
**Presentation:** 3
**Contribution:** 3
**Rating:** 5
**Confidence:** 5

**Summary:**

This is an empirincal paper. The paper presents MEGALODON, a neural architecture designed to overcome the quadratic complexity and weak length extrapolation of Transformers. By extensive experiments, this  paper demonstrates MEGALODON's ability to efficiently handle unlimited context lengths and its superior performance across different tasks and modalities. The claimed key contributions of this paper is as follows::
1) improves upon the MEGA architecture by adding the complex exponential moving average (CEMA), timestep normalization, normalized attention, and a pre-norm with two-hop residuals.
2. achieves better pretraining efficiency and downstream task accuracy than Transformers, specifically in handling long sequences with 7 billion parameters and 2 trillion training tokens.
3. outperforms Transformers across various benchmarks, including long-document comprehension, multi-turn conversation, and video generation.

**Strengths:**

1. The performance of the proposed architecture is excellent, as demonstrated in Table 1.
2. Extensive experiments are conducted to evaluate the proposed methods.
3. The topic of this paper is important and critical for the LLM domain.

**Weaknesses:**

1. The training curves of MEGALODON and LLAMA2 7B in Figure 1 cross at 750 billion training tokens. It would be interesting to see if they cross again with more tokens, such as at 6 trillion.
2. Section 3.2 is difficult to understand due to the lack of background and intuitive explanations.
3. Section 3.5 makes overclaims about 4D parallelism. This topic is well-explored and relevant, as discussed in [1, 2, 3], but these references are ignored. If the paper aims to highlight the benefits of 4D parallelism, it should include comparative experiments.
4. The code link provided in the abstract does not work.
5. The long context tasks in Section 4.3, as shown in Tables 2 and 6, are not fairly set up. In Table 2, the only fair baseline is LLAMA2-L, which performs better. This raises doubts about the proposed method's long-context ability since other models (Yarn, MPT, Xgen) use different training datasets, potentially limiting their long-context capability. For Table 5, the baselines (Transformer, Reformer, ...) also use different training datasets, making the comparisons misleading. Compared to MEGA, the improvements are marginal. Thue the compression of the long-ctx ability is not convincing to me.


[1] Sequence Parallelism: Making 4D Parallelism Possible\
[2] Lightseq: Sequence level parallelism for distributed training of long context transformers\
[3] USP: A Unified Sequence Parallelism Approach for Long Context Generative AI

**Questions:**

see Weaknesses

**Limitations:**

see Weaknesses

---

> ### Author Rebuttal · Authors · 2024-08-06
>
> Thanks for your time and constructive comments! We appreciate your positive feedback on the good motivation, novelty of Megalodon and its strong empirical results. We address your concerns and questions below and please let us know if you still have concerns after you read our response.
>
> > W1: The training curves of MEGALODON and LLAMA2 7B in Figure 1 cross at 750 billion training tokens. It would be interesting to see if they cross again with more tokens, such as at 6 trillion.
>
> We did not have sufficient computational resources to compare Megalodon and Transformer on a scale of 6T training tokens. But based on the trends of the learning curves in Figure 1, we believe that the gap between Megalodon and Transformer might not be narrowed with more training data.
>
> > W2: Section 3.2 is difficult to understand due to the lack of background and intuitive explanations.
>
> As discussed in the beginning of Section 3.2, the motivation of Timestep Normalization is to perform normalization along the axis of time steps, which serves the motivation of reducing internal covariate shift along the spatial dimension, similar to Batch Normalization and Group Normalization. The effectiveness of Timestep Normalization has been shown in the [author rebuttal](https://openreview.net/forum?id=XlAbMZu4Bo&noteId=G72sD1BBxi).
>
> > W3: Section 3.5 makes overclaims about 4D parallelism. This topic is well-explored and relevant, as discussed in [1, 2, 3], but these references are ignored. If the paper aims to highlight the benefits of 4D parallelism, it should include comparative experiments.
>
> Thanks for pointing out the related work we missed. In fact, we have cited [2] in our submission. We will add the other two works in our final version.
>
> In Section 3.5, we want to highlight the benefit of chunk-wise attention in parallel pre-training. As explored in [1-4], advanced algorithms have been developed for distributed computation of full attention. However, these algorithms involves significant communication costs. Benefiting from the chunk-wise attention in Megalodon, there are no communications of attention along the chunk/context parallel groups.
>
> [1] Sequence Parallelism: Making 4D Parallelism Possible
>
> [2] Lightseq: Sequence level parallelism for distributed training of long context transformers
>
> [3] USP: A Unified Sequence Parallelism Approach for Long Context Generative AI
>
> [4] Ring Attention with Blockwise Transformers for Near-Infinite Context
>
> > W4: The code link provided in the abstract does not work.
>
> The anonymous link is a placeholder, not a real link. We will release the code in the final version.
>
> >W5: The long context tasks in Section 4.3, as shown in Tables 2 and 6, are not fairly set up. In Table 2, the only fair baseline is LLAMA2-L, which performs better. For Table 5, the baselines (Transformer, Reformer, ...) also use different training datasets, making the comparisons misleading. Compared to MEGA, the improvements are marginal. Thue the compression of the long-ctx ability is not convincing to me.
>
> In fact, the model that was trained on the same data with Megalodon is LlaMa2, not LlaMa2-L.  LlaMa2-L continually trained LlaMa2 on 500B selected long-context data for length extension. That is the partial reason why LlaMa2-L performs slightly better in Table 2.
> For other long-context models, such as Yarn, Xgen and MPT, we agree that they are trained on different data and the comparison is not entirely fair. However, it is impractical to re-train all these models with the same data.
>
> For the results in Table 6 on LRA, all the models (Transformer, Reformer, etc.) are trained on the same datasets for each task. Thus, the comparison in Table 6 is fair.
>
> When we compare Megalodon-chunk with Mega-chunk in Table 6, the improvements are significant (87.62 vs. 85.66).
> In Table 6, the improvements of Megalodon over Mega with full attention are less significant than that of the chunk-wise attention models. It is reasonable because the benefits of Megalodon over Mega might be covered by the full attention mechanism.

---

### Official Review · Reviewer_U1i1 · 2024-07-17

**Soundness:** 3
**Presentation:** 3
**Contribution:** 3
**Rating:** 6
**Confidence:** 3

**Summary:**

The paper proposes Megalodon, which introduces three advancements over Mega: complex EMA, timestep normalization, and normalized attention. These advancements address the limitations of chunk-wise attention and architecture divergence across different tasks and data types. The new model is evaluated alongside Llama-2, both trained on the same public dataset, and demonstrates competitive and superior performance across a wide range of benchmarks.

**Strengths:**

1. Clear motivations: All three improvements directly target the limitations of Mega.
2. The complex EMA is a novel approach.
3. The authors provide efficient parallelism.

**Weaknesses:**

Recent theoretical work [1] has shown that efficient versions of attention (like the chunk-based method used in this paper) can limit the expressiveness of the model, particularly for reasoning tasks that involve long-range information. The paper evaluates Megalodon with long-context open-book QA tasks. How will Megalodon perform on PhoneBook lookup [2] with ICL, especially for phonebook lengths longer than 4K tokens? How will it perform on complex reasoning tasks requiring long context, such as 8-shot or 16-shot math problems with GSM8K or coding tasks on HumanEval?

Empirically, it is not clear how CEMA improves expressiveness. It would be most direct to compare using CEMA versus using EMA on these tasks.

The reviewer understands that the rebuttal period is short and is therefore not requiring most of these experiments to be added.

[1] Yang, Kai, et al. "Do Efficient Transformers Really Save Computation?" Forty-first International Conference on Machine Learning.

[2] Jelassi, Samy, David Brandfonbrener, and Sham M. Kakade. "Repeat After Me: Transformers are Better than State Space Models at Copying." Forty-first International Conference on Machine Learning.

**Questions:**

Have the authors conducted ablation studies for small-scale models before moving to 7B, similar to the results in Appendix C? Including these ablation studies, if already performed, would help readers understand how the three designs impact performance.

**Limitations:**

The paper does not discuss its limitations. The authors believe there is no negative societal impact. In the paper checklist, justifications are required for answers marked "Yes," but the authors have deleted the justification for several items. For limitations, the authors claim they are discussed, but there is no justification provided.

The anonymous link is not working.

---

> ### Author Rebuttal · Authors · 2024-08-06
>
> Thanks for your time and constructive comments! We appreciate your positive feedback on the good motivation, novelty of Megalodon and its strong empirical results. We address your concerns and questions below and please let us know if you still have concerns after you read our response.
> > W1 & Q1: Have the authors conducted ablation studies for small-scale models before moving to 7B, similar to the results in Appendix C?
>
> The ablation studies on small-scale benchmarks are in the [author rebuttal](https://openreview.net/forum?id=XlAbMZu4Bo&noteId=G72sD1BBxi).
>
> > W2:  Recent theoretical work [1] has shown that efficient versions of attention can limit the expressiveness of the model…
>
> We conducted experiments to evaluate Megalodon on retrieval-oriented tasks, such as passkey retrieval. Similar to previous studies, due to chunk-wise attention, Megalodon under-performed on these retrieval-oriented tasks compared with full attention mechanism, but outperformed state-space models such as Mamba.
>
> For example, without any fine-tuning for length extension, Megalodon completes the passkey retrieval task with up to 16K context length, while Mamba can only complete this task up to 4K context. Long-LlaMa2, which continually trains LlaMa2 on selected long-context data for length extension, is able to complete up to 32K context length.
>
> Further improving Megalodon for retrieval-oriented tasks is an interesting and important direction for future work.

---

> > ### Comment · Reviewer_U1i1 · 2024-08-09
> >
> > Thanks for the rebuttal and additional ablation studies. After reviewing the new information, I am maintaining my original scores and continue to recommend acceptance.

---

### Author Rebuttal · Authors · 2024-08-06

# Ablation studies on CEMA and Timestep Normalization

**Ablation on LRA**

We first conducted ablation studies on LRA to demonstrate the effectiveness of CEMA and Timestep Normalization components in Megalodon. The results are shown in the following table:

| Models                    |  ListOps (LN)     |  Text (SN)     |  Retrieval  (SN)    |  Image (BN)     |  Pathfinder (BN)  | Path-X (BN) |  Avg. |
| :--------- | :----------------: | :----------------: | :----------------: | :----------------: |  :----------------: | :----------------: | :----------------: |
| Mega (EMA)                     |   58.76   |   90.19   |   90.97   |   85.80   |   94.41   |   93.81   |   85.66   |
| Megalodon (CEMA)          |   61.13   |   90.58   |   91.51   |   87.32   |   96.11   |   96.98   |   87.37   |
| Megalodon (CEMA&TSN)|  62.25   |   90.50   |   91.76   |   87.16   |   96.85   |   97.21   |   87.63   |

In this study, we used the Mega architecture as the baseline, which uses different normalization layers for different tasks in LRA. The normalization layers for different tasks are in the brackets following the task names in the table headline. BN, LN and SN refer to Batch Normalization, Layer Normalization and Scale Normalization.

The second row in the above table is the Megalodon architecture by replacing EMA with CEMA. For this architecture, we used the same normalization layers with the original Mega on different tasks. The third row is the Megalodon architecture with both CEMA and Timestep Normalization (TSN). Note that for this architecture, we used the same TSN for all the six tasks in LRA.

**Ablation on WikiText-103**

We then conducted ablation studies on auto-regressive language modeling on the moderate WikiText-103 dataset:

| Models                              |    #Param.   |     PPL    |
| :-----------------------------| :------------: | :--------: |
| Mega (EMA&LN)               |       252M     |   18.07   |
| Megalodon (CEMA&LN)   |       252M     |    17.63  |
| Megalodon (CEMA&TSN) |       252M    |    17.23  |

In this study, we also used the Mega architecture with EMA and Layer Normalization (LN) as baseline. The second row is Megalodon with CEMA and Layer Normalization, while the third row is the Megalodon with CEMA and Timestep Normalization (TSN).

From the two ablation studies, both CEMA and Timestep Normalization show improvements over the original Mega architecture.


# Effect of normalized attention, and pre-norm with two-hop residuals

The normalized attention and pre-norm with two-hop residuals are designed to improve stability of Megalodon in large-scale pre-training. Their effects on small-scale models are not significant.

---

### Decision · Program_Chairs · 2024-09-25

**Decision:**

Accept (poster)

**Comment:**

This paper addresses long context length issue by introducing new architecture.

Intially, three reviewers voted acceptance and one reviewer gave a borderline rejection with lack of experiments.

However, the authors successfuly addresss main concerns that reviewers raised, thus all reviewers gave positive scores.

AC also agree with the reviewers opinion, and moreover AC would like to encourage research on new architecture apporach for topic diversity in LLM regimes.

So, AC recommens accepting this paper.